# Neoadjuvant PD-1 blockade induces T cell and cDC1 activation but fails to overcome the immunosuppressive tumor associated macrophages in recurrent glioblastoma

Alexander H. Lee [1,2,8], Lu Sun[1,8], Aaron Y. Mochizuki [1], Jeremy G. Reynoso [1], Joey Orpilla [1], Frances Chow [3], Jenny C. Kienzler [1], Richard G. Everson [1,4], David A. Nathanson[2,4], Steven J. Bensinger [2,4,5], Linda M. Liau [1,2,4], Timothy Cloughesy [2,3,4], Willy Hugo[4,6,7✉] & Robert M. Prins [1,2,4,6✉]

Primary brain tumors, such as glioblastoma (GBM), are remarkably resistant to immunotherapy, even though pre-clinical models suggest effectiveness. To understand this better in patients, here we take advantage of our recent neoadjuvant treatment paradigm to map the infiltrating immune cell landscape of GBM and how this is altered following PD-1 checkpoint blockade using high dimensional proteomics, single cell transcriptomics, and quantitative multiplex immunofluorescence. Neoadjuvant PD-1 blockade increases T cell infiltration and the proportion of a progenitor exhausted population of T cells found within the tumor. We identify an early activated and clonally expanded CD8+ T cell cluster whose TCR overlaps with a CD8+ PBMC population. Distinct changes are also observed in conventional type 1 dendritic cells that may facilitate T cell recruitment. Macrophages and monocytes still constitute the majority of infiltrating immune cells, even after anti-PD-1 therapy. Interferon-mediated changes in the myeloid population are consistently observed following PD-1 blockade; these also mediate an increase in chemotactic factors that recruit T cells. However, sustained high expression of T-cell-suppressive checkpoints in these myeloid cells continue to prevent the optimal activation of the tumor infiltrating T cells. Therefore, future immunotherapeutic strategies may need to incorporate the targeting of these cells for clinical benefit.

[1] Department of Neurosurgery, University of California, Los Angeles, Los Angeles, CA 90095, USA. [2] Department of Molecular and Medical Pharmacology, University of California, Los Angeles, Los Angeles, CA 90095, USA. [3] Department of Neurology/Neuro-Oncology, University of California, Los Angeles, Los Angeles, CA 90095, USA. [4] UCLA Jonsson Comprehensive Cancer Center, University of California, Los Angeles, Los Angeles, CA 90095, USA. [5] Department of Microbiology, Immunology, and Molecular Genetics, University of California, Los Angeles, Los Angeles, CA 90095, USA. [6] Parker Institute for Cancer Immunotherapy, 1 Letterman Drive, Suite D3500, San Francisco, CA 94129, USA. [7] Department of Medicine/Dermatology, University of California, Los Angeles, Los Angeles, CA 90095, USA. [8]These authors contributed equally: Alexander H. Lee, Lu Sun. ✉email: HWilly@mednet.ucla.edu; RPrins@mednet.ucla.edu

Recurrent glioblastoma is associated with a median overall survival of 24–44 weeks[1–3]. Although immunotherapy such as checkpoint blockade has revolutionized the treatment of several cancers, its benefit in glioblastoma (GBM) has been limited to small randomized trials in the neoadjuvant setting[4], with no FDA approval thus far.

In other cancer types, characterization of the immune composition of the tumor microenvironment has revealed significant heterogeneity across tumor subtypes and patients, with high diversity in the intratumoral compartments[5–10]. In certain cases, unique novel lymphoid or myeloid subsets emerge within the tumor that are absent from adjacent normal tissue or peripheral blood, and such specific immune compositions may affect survival. In the GBM tumor microenvironment, myeloid cells are known to comprise a significant proportion of immune cells[5,11–17].

We previously demonstrated that neoadjuvant anti-PD-1 therapy (neo-aPD1) was associated with improved overall and progression-free survival in a small randomized phase 1 clinical trial[4]. Treatment with PD-1 blockade prior to surgical debulking induced a significant increase in interferon-γ-related gene expression and an associated decrease in cell cycle-related gene expression. However, this approach was performed on bulk tissue and did not capture the perturbations in specific cell types within the tumor microenvironment, and although neo-aPD1 was associated with a significant survival benefit, it was not curative, suggesting that there are other interactions within the microenvironment that prevent neo-aPD1 from being completely effective.

The effect of neo-aPD1 is believed to be predominantly driven by re-activation of exhausted T cells. However, PD-1 blockade may also shape the tumor microenvironment and influence non-T cell populations[18,19]. One publication performed high dimensional analysis of the effect of checkpoint blockade in altering the immune landscape of GBM;[20] however, this study was limited to a small number of patients. Another recent publication examined the tumor myeloid and dendritic cell populations of patients with newly diagnosed GBM or recurrent GBM; three of the latter were treated with neo-aPD1 24 h prior to surgery[21]. The authors noted that this would be too short of a timeframe for the neo-aPD1 treatment to meaningfully affect the immune composition of the tumor microenvironment. Therefore, the abundance and differentiation of T cell and myeloid cell subsets after the administration of neo-aPD1 in the GBM setting is not well characterized and additional work is needed to provide a more comprehensive understanding of the effects of immunotherapy in GBM. To address this question, here we use time-of-flight mass cytometry (CyTOF) and/or single-cell RNA sequencing (scRNAseq) to analyze the tumor-infiltrating CD45+ immune cell population at the single-cell level of 69 GBM patients who underwent surgical resection at the University of California, Los Angeles.

Neo-aPD1 significantly increases the proportion and number of T cells in the tumor microenvironment but the immune microenvironment remains largely dominated by myeloid cells. Single-cell RNA and TCR sequencing analysis suggests that neo-aPD1 expands an early activated, cytotoxic CD8 T cell population in the peripheral blood that traffics into the tumor microenvironment and produces a population of progenitor exhausted CD8 T cells that has been previously identified in the chronic viral setting[22–24] and in melanoma[25].

Our scRNAseq analysis also shows that the T cell populations activated by neo-aPD1 produces chemotactic factors that recruit DCs (*XCL1*, *XCL2*) and additional T cells (*CCL5*) into the tumor microenvironment. Newly trafficked T cells from the periphery expand but eventually transition into a progenitor exhausted state, potentially through the engagement

of the TIGIT and CTLA-4 immune checkpoints on these cells. In the myeloid population, neo-aPD1 is associated with IFN-driven transcriptional changes, leading to some myeloid populations that secrete T cell chemotactic factors such as CXCL9/10/11. However, the overall myeloid population is still dominated by multiple tumor-associated macrophage (TAM) populations which highly express *NECTIN2* and *CD86*, the genes encoding the interaction partner of TIGIT and CTLA-4. Indeed, the interactome analysis based on the single-cell gene expression indicates that neoadjuvant anti-PD-1 is associated with increased interactions between the T cell checkpoints TIGIT and CTLA4 on T cell populations and their respective receptors in the myeloid populations. Moreover, neo-aPD1 induces a CXCR4+ TAM population and does not reduce multiple myeloid populations that are often associated immune suppression.

Collectively, in this work, we show that treatment with neo-aPD1 remodels the cellular immune composition of the GBM tumor microenvironment across multiple immunologic nodes. Despite the alterations in the tumor-infiltrating lymphocyte (TIL) and cDC populations following neoadjuvant checkpoint blockade, T-cell-suppressive myeloid cells still dominate the immune landscape of these tumors. Additional strategies targeting TIGIT and/or CTLA-4 may be needed to improve the strength and durability of antitumor T cell response in neo-aPD1 treated GBM patients.

## Results

**Patient characteristics, data acquisition, and initial data analysis of all immune populations.** To understand how neo-aPD1 changes the immune landscape of the tumor microenvironment, we used CyTOF mass cytometry (CyTOF), single-cell RNA sequencing (scRNAseq), and/or multiplex immunofluorescence (mIF) to analyze the tumor-infiltrating CD45+ immune cells from a total of 69 unique GBM patients (Fig. 1a, b). At surgery, 27 of these patients had newly diagnosed GBMs (GBM.new), 22 of these patients had recurrent GBM without prior immunotherapy (GBM.rec), and 20 of these patients had recurrent GBM with neo-aPD1 therapy (GBM.pembro). In all, 34 of the patients had tumor-infiltrating immune cells that were analyzed with both CyTOF and scRNAseq, 29 patients with CyTOF alone, and 6 patients with scRNAseq alone. We also analyzed PBMCs from 5 GBM.rec, 8 GBM.pembro, and 2 healthy donor patients using scRNAseq. All GBM patients whose PBMCs were analyzed also had their tumor-infiltrating immune cells analyzed with scRNAseq. We collected tumor sections for 20 patients (8 GBM.rec and 12 GBM.pembro) to stain with multiplex immunofluorescence, 5 of whom were also analyzed with either CyTOF and/or scRNAseq.

For the initial CyTOF analysis, we sampled up to 20,000 of the live cells per patient ($n = 1,056,057$ total cells) and performed unbiased clustering using ClusterX[26] based on the expression of 22 cell surface markers (Supp. Data 1) across our 63 patient cohort. The tumor-infiltrating immune cell clusters were grouped into six populations (Fig. 1c, d and Supp. Fig. 1a–c): T cells (CD3+), myelo-monocytic cells (CD14/CD33+), granulocyte (CD15+), natural killer cells (CD56+ and CD16+), and two uncharacterized populations (CD56+ and no markers). We also analyzed tumor-infiltrating immune cells from 40 GBM patients ($n = 156,766$ cells) using scRNAseq. We selected the clustering resolution that separated the cells into six clusters (Fig. 1e): one cluster of lymphoid cells based off expression of *CD3D*, two clusters of myeloid cells based off expression of *CD14* and *FCGR3A/CD16*, one cluster of stressed/dying cells (dominated by long non-coding RNAs like *MALAT1, NEAT1,*

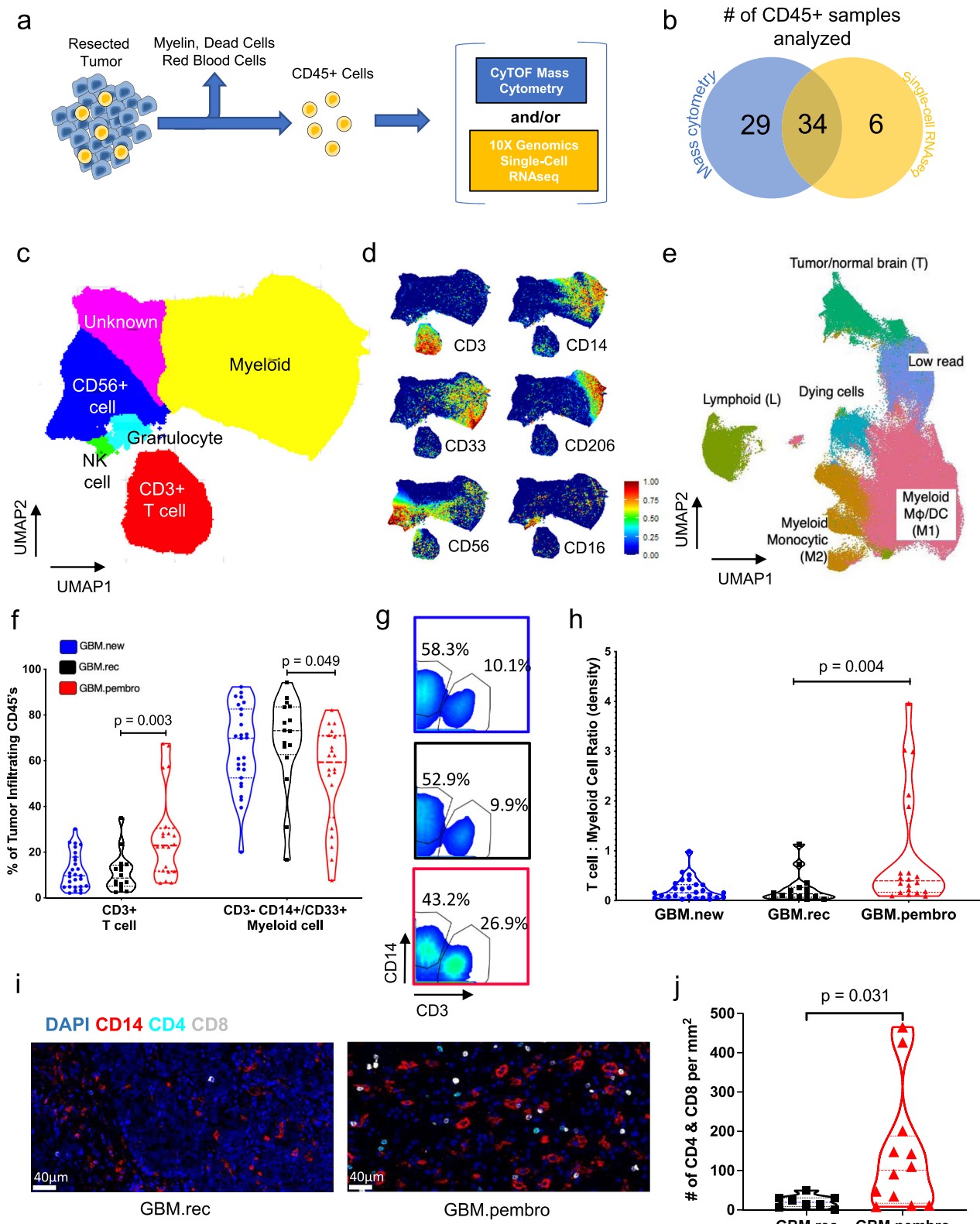

and heat shock proteins (HSPs)), one cluster of tumor or normal brain cells based off expression of *GFAP* and *SOX2*, and one cluster of cells that had too few reads to accurately classify (Supp. Fig. 1d and Supp. Data 2). It is possible that the CyTOF clusters that had little marker expression or only CD56 expression also correspond to the tumor cluster from our scRNAseq analysis.

**Neo-aPD1 therapy increases the overall T cell infiltrate in the GBM microenvironment.** We began our analysis looking at the overall changes to the tumor-infiltrating T cell and myeloid cell compartments after neo-aPD1. Our CyTOF analysis showed a significant increase in the proportion of CD3+ T cells in the tumor-infiltrating immune populations in the GBM.pembro group and a corresponding decrease in the myelo-monocytic

**Fig. 1 Neo-aPD1 increases the proportion and number of CD3+ T cells among tumor-infiltrating immune cells in recurrent glioblastoma patients.**
**a** Schematic of CD45+ cell isolation from tumor tissue. **b** The number of patients whose tumor-infiltrating CD45+ cells were analyzed using CyTOF and/or scRNAseq. **c** A UMAP projection of the CD45+ tumor-infiltrating cells analyzed with CyTOF on the common 22 marker panel (Supp. Data 1, $n = 1,067,057$ cells from 63 total patients: 26 GBM.new, 17 GBM.rec, 20 GBM.pembro). **d** Select marker intensities on the UMAP projections of all CD45+ cells. **e** A UMAP projection of CD45+ tumor-infiltrating cells analyzed with scRNAseq ($n = 156,766$ cells from 40 total patients: 14 GBM.new, 12 GBM.rec, and 14 GBM.pembro). **f** The percentage of tumor-infiltrating CD45+ cells belonging to T and myeloid cell clusters from (**c**) across different tumor groups (blue circles: GBM.new, black squares: GBM.rec, red triangles: GBM.pembro; each dot represents a different patient). **g** CD3 and CD14 gating in the CyTOF data separated by different tumor groups. Colors are the same as **f**. **h** The number of CD3+ T cells per mg of tumor across different tumor groups. The highest value for each condition was treated as outlier and removed. Colors are the same as **f**. **i** Representative 20x images of multiplex immunofluorescence staining of a GBM.rec (left) and GBM.pembro (right) tumor samples (blue: DAPI, red: CD14, cyan: CD4, white: CD8). Images were collected and analyzed for 8 GBM.rec and 12 GBM.pembro patients. **j** The number of CD4+/CD8+ cells per mm$^2$ of tissue section for GBM.rec (black squares) and GBM.pembro (red triangles) samples. $P$ values were calculated using a two-sided Wilcoxon rank-sum test. Source data are provided as a Source Data file or in Supp. Data 2.

populations compared to the GBM.rec groups (Fig. 1f, g). Our scRNAseq analysis showed a similar trend towards decreased proportion of the myeloid population (Supp. Fig. 1e and Supp. Data 2).

To evaluate whether the increased proportion of T cells was caused by increased T cell infiltration into the tumor and not by loss of myeloid cells in the tumor, we calculated the total number of the T cells and myelo-monocytic cells per mg of tumor dissociated. The absolute number of T cells per mg of tumor significantly increased in GBM.pembro compared to GBM.new and GBM.rec (Fig. 1h), while the number of myeloid cells remained relatively constant (Supp Fig. 1f). In parallel, our multiplex immunofluorescent staining also showed a concordant increase in the number of CD4+ and CD8+ T cells per mm$^2$ of tumor section in GBM.pembro patients compared to GBM.rec patients (Fig. 1i, j), while the number of CD14+ myelo-monocytic cells was similar (Supp Fig. 1g). In summary, by using multiple approaches, we showed that neo-aPD1 is associated with increased relative proportion and absolute total number of T cells in the tumor microenvironment.

**Neo-aPD1 activates T cells that produce chemotactic and cytolytic factors and induced the generation of the progenitors of exhausted T cells.** Recent work in the chronic viral setting has shown that there are different stages of T cell activation and exhaustion[22–25]. To analyze the lymphoid compartment in finer detail than our CyTOF analysis and to see whether we could detect these differentially activated/exhausted T cell subsets, we increased the clustering resolution of the lymphoid cells from our initial scRNAseq clustering ($n = 14,322$). There were 11 resulting clusters that included: four T cell clusters (effector CD8 L1: *CD8A, GZMK, CCL5*; early activated L2: *IL7R, CD40LG, CD69*; effector with progenitor phenotype L3: *TCF7, CCR7, IL7R, GZMK/B, PRF1, CTLA4, SLAMF6*; cytolytic T cluster L5: *XCL1/2, GZMA/B, KLRB1, PRF1*), one T$_{reg}$ cluster (L4: *FOXP3, CTLA4*), one cluster of proliferating lymphoid cells (L6: *MKI67, CDK1*), one pDC and B cell cluster (L7: *CD79A, LILRA4*), three clusters with both lymphoid and myeloid signatures (L+M1-3: *CD14, CD68, C1QA, TREM2, GPNMB*), and one cluster of dying/stressed lymphoid cells (D: *NEAT1*) (Fig. 2a, Supp. Fig. 2a, and Supp. Data 3).

Among the T cells clusters (L1-5), there was an increased proportion of *GZMK+ CCL5+ CD8+* T cell (L1), T$_{reg}$ population (L4), and a smaller increase in the progenitor-like effector cells (L3) (Supp. Fig. 2b and Supp. Data 3). Of note, genes and gene sets related to IFN-γ activation (*STAT1, IRF1, CXCL9*, and *GBP4*) and T cell activation and exhaustion (*CXCL13, ICOS, CTLA4, PDCD1, TOX, BATF, LAG3, TIGIT, CD226*, and *SLAMF6*) were significantly upregulated in response to neo-aPD1 (Fig. 2b and

Supp. Fig. 2c), likely driven by the overall increase of *IFNG* expression in the GBM.pembro samples (Fig. 2c). The overall increase in *IFNG* levels also correlated with the enrichment of the interferon downstream genes in the lymphoid cells, indicating an actual activation of the interferon pathway (Supp. Fig. 2d). Protein levels of *ICOS, TIGIT*, and *TNFRSF9*, which were covered in our expanded 34 marker CyTOF panel (applied on a subset of 35 patients from our previous CyTOF analysis; 14 GBM.new, 7 GBM.rec, 14 GBM.pembro; $n = 565,654$ cells, Supp. Data 1), also showed a trend towards increased surface expression upon neo-aPD1 treatment (Supp. Fig. 2e, f and Supp. Data 1).

Next, we performed a pseudotime trajectory analysis using Monocle 2[27] on CD4 and CD8 T cells from the four T cell clusters: L1, L2, L3, and L5 (cluster L6 of proliferating T cells was excluded because it was dominated by cell cycle genes). In the CD4 T cell compartment (Fig. 2d, e), neo-aPD1 significantly increased the proportion of proliferating, intermediate-exhausted Th1 T cells (Th1 exh state: *TCF7, CCR7, BACH2, TBX21, IFNG, STAT1, IRF1, MKI67, TOX, ENTPD1*) compared to GBM.rec. There was also a smaller increase ($\geq$ 1.5-fold) in the early effector CD4 population (Th0 Ac: *CD69, CCL5*) and the IFNγ-activated, cytotoxic Th17-like population (Th17 memory-1: *RORC, GZMB, STAT1, IRF1, CXCL9*), also showing the expression of progenitor gene markers (*TCF7, CCR7, IL7R*).

In the CD8 T cell compartment (Fig. 2f, g and Supp. Data 3), neo-aPD1 increased the proportion of IFNγ-activated, progenitor exhausted T cells. This population upregulated both cell cycle and cytolytic genes (Exh-prog: *TCF7, CCR7, IL7R, BACH2, GZMB, PRF1, CXCL9, STAT1, IRF1, MKI67, TBX21, SLAMF6*). Phenotypically, this proliferating and cytolytic *TCF7+* population also expressed *SLAMF6*, closely matching the Tex$^{prog2}$ and Tex$^{int}$ populations of early progenitor of exhausted T cells recently described by Beltra et al[23]. Correlating the fates back to the Seurat's L1-L5 clusters, we noted that the CD8 T cells in this fate were dominated by those from the L3 cluster (Supp. Fig. 2g), indicating that the majority of the L3 T cells are in this progenitor exhausted T cell state.

Interestingly, the three *GZMK+* effector subsets (Eff 4, 5, and 6, dominated by L1 and L5 in the Seurat analysis) expressed *KLRB1* and *XCL1*. This is consistent with the recent findings on glioma-infiltrating T cells[28]. Among these three subsets, Eff 5 and 6 expressed high *PDCD1* and *CCL5*, compared to Eff 4. Eff 6, whose proportion was induced by neo-aPD1, specifically expressed high levels of *HAVCR2* and *IFNG*, implying activated, antigen-specific CD8+ T cells. Reassuringly, both *CCL5* and *XCL1* were higher after neo-aPD1 when we look across the total cells (Fig. 2h). XCL1 and CCL5 are factors that have been reported to attract dendritic cells into the tumor microenvironment[29] or during viral infection[30].

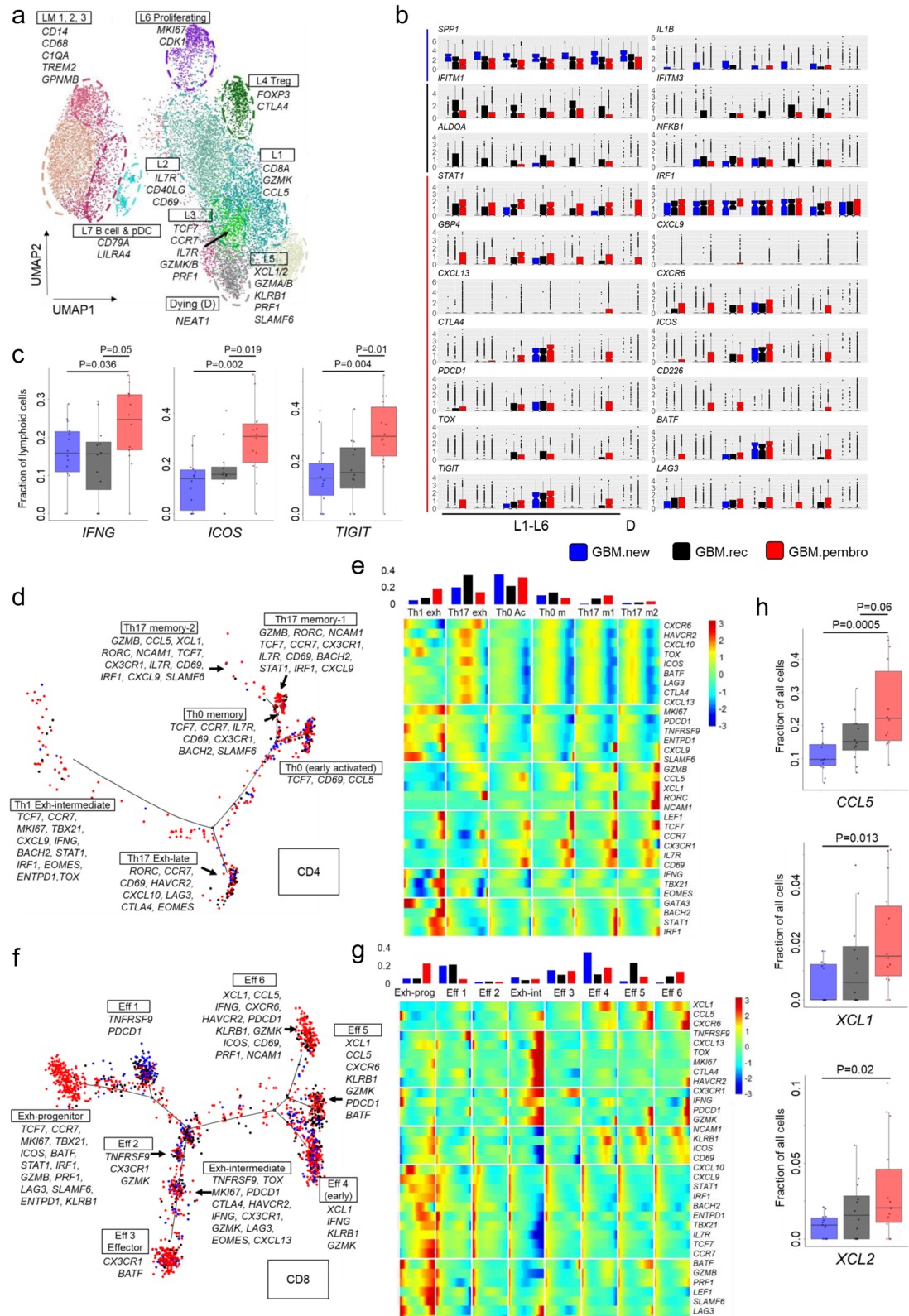

In all, neo-aPD1 treatment induced the expansion and proliferation of a progenitor-like, exhausted T cell population upregulating IFN-γ, T-bet, and cytolytic markers, indicating that they may be the tumor antigen-specific T cells. Additionally, neo-aPD1 also induced populations of effector T cells that upregulate the secretion of chemotactic factors associated with DC recruitment.

**TCR clone analyses demonstrates shared clones among the cytolytic T cells in the peripheral blood and intratumoral T cells**. To evaluate whether we could detect early changes in the systemic circulation with neo-aPD1, we analyzed the scRNAseq data of the PBMC samples from a subset of the GBM.rec and GBM.pembro patients. We clustered PBMC lymphoid cells (n = 56,444 cells) into 15 different clusters of T cells, NK cells,

**Fig. 2 Single-cell RNAseq analysis of intratumoral T cells shows transcriptional changes with neo-aPD1 therapy. a** A UMAP projection of the lymphoid compartment of the tumor samples analyzed using scRNAseq ($n = 14,322$ cells from 40 total patients: 14 GBM.new, 12 GBM.rec, and 14 GBM.pembro). **b** Differentially expressed genes in GBM.new (blue), GBM.rec (black), and GBM.pembro (red) patient samples as computed by Seurat. **c** The fraction of lymphoid cells with detected expression of the indicated genes. **d**, **f** Pseudotime projection of CD4 (**d**) and CD8 (**f**) T cells by Monocle 2. **e**, **g** The expression of CD4 (**e**) and CD8 (**g**) T cell marker genes across the different cell fates in **d** and **f**, respectively. (top) Proportion of the cells from in each fate, separated by tumor groups, are shown. **h** The percentage of all cells expressing at least one transcript for *CCL5* and *XCL1/2*. *P* values were calculated using a two-sided Wilcoxon rank-sum test. Source data are provided as a Source Data file or in Supp. Data 3. In all boxplots, the median is indicated by the line within the box and the 25th and 75th percentiles indicated by the lower and upper bounds of the box. The upper and lower lines above and below the boxes represent the whiskers.

and B cells (Fig. 3a, Supp. Fig. 3, and Supp. Data 4). There was one cluster of lymphoid and myeloid doublets and one cluster enriched with blood platelet related mRNA. Neo-aPD1 induced an increase in the largest cluster of cytolytic T/NK cell population in the PBMC (P3) (Fig. 3b) and significantly upregulated genes that are related to cell proliferation across all clusters (Fig. 3c and Supp. Data 4). This indicated that neo-aPD1 induces a systemic activation and expansion of T cell clones in the periphery.

To detect the CDR3 regions of the T cells in our 3′ scRNAseq data, we utilized an algorithm called TRUST4[31] to define distinct TCR clones by the resolved CDR3 sequences of the TCRβ chain of a T cell. We observed that the T cells with expanded clones (clone size >1) tended to be within the early effector *GZMK+* (L1 in the TIL and P2 in the PBMC) and cytolytic *GZMB+* clusters (L5 in TIL and P1, P3 in PBMC) (Fig. 3d and Supp. Data 5).

Next, we used STARTRAC analysis[32] to examine the pattern of overlaps of these major TCR clones in the different T cell clusters. In the PBMC compartment, the highest transition score was between the *GZMK+* P2 cluster and the *GZMB/H+* P1 cluster. Both P1 and P2 also showed a higher transition score with the cytolytic cluster P3, which suggests that both of these early effector populations in P1 and P2 may differentiate to the cytolytic, effector T cells in P3 (Fig. 3e).

Comparing between PBMC and TIL, we observed that the TCRs of the cytotoxic TILs in L5 overlapped with PBMC's P1-4, with the strongest overlap between the cytotoxic clusters of the TIL (L5) and PBMC (P3) (Fig. 3f). There was also an overlap between the *GZMK+* clusters of both the TIL and PBMC (L1 and P1-2, respectively). This observation suggests that the intratumoral cytolytic T cells in L5 are populated by the PBMC's early effector CD8 T cells (P1, P2), the cytolytic T cells (P3), and the proliferating T cells (P4). Notably, a recent publication reported that peripheral expansion of certain T cell clonotypes correlates with the infiltration of those clonotypes into the tumor[33]. In the TIL compartment, the highest transition score was observed between the L3 and L5 clusters, suggesting a transition from a cytolytic, effector phenotype to a progenitor exhausted phenotype (Fig. 3g), which was enriched after neo-aPD1.

The observation of higher clonal frequencies of the cytolytic T cells within the tumor (*GZMB+*, *PRF1+*, and *KLRB1+*) was reported recently by Matthewson et al.[28], and our results highlight the systemic source of such cytolytic T cells and how neo-aPD1 enhances its clonal expansion. Most of the expanded cytolytic T cell clones in the PBMC probably involves existing large clones that recognize pathogens, of which some may be cross reactive to tumor antigens[34]. Meanwhile, the progenitor exhausted T cells in L3 only showed clonal overlaps with those in intratumoral cytolytic T cells in L5 but not with any of the clusters in the PBMC. This implies that the T cell clones in L3 were present at much lower clonal frequencies in the PBMC (below the detection level of TRUST4) and that they subsequently became activated and then clonally expanded only after their encounter with tumor antigens.

**Neo-aPD1 changes the composition and transcriptional profile of myelo-monocytic cells, leading to increased T cell trafficking and immunosuppressive activity.** Because our previous published work indicated that neo-aPD1 led to an increased T cell and interferon signature, we interrogated whether this interferon signature may affect the proportion or function of other non-lymphoid immune populations.

Our scRNAseq subclustering of the myeloid cells ($n = 72,492$ cells) revealed 11 clusters that included: six macrophage clusters (Mφ1: *IL1B, CCL3/4*; Mφ2: *CXCR4, MHC II*; Mφ3: *CXCL10, MHC II*; Mφ4: HSP high, Mφ5: *TMEM119, ADORA3*, microglia-enriched, Mφ6: *GPNMB, CSTB*; and Mφ7: *MRC1, ANGPTL4*), one monocyte cluster (M: *FCN1,VCAN, S100A8/9*), one DC cluster (DC: *FCER1A, MHC II*), one cluster with both lymphoid and myeloid signatures (L+M: *CD3D*), and one proliferating myeloid cell cluster (P: *MKI67, CDK1*) (Fig. 4a, b and Supp. Data 6).

Mφ1-5 populations were very similar where Mφ1 showed a higher expression of macrophage and monocyte inflammatory factors *CCL3, CCL4, IL1B*, and *CXCL8*, Mφ2 showed higher expression of *CXCR4* and *MHCII* transcripts, Mφ3 had increased interferon stimulated genes (ISGs), Mφ4 showed higher proportion of heat shock proteins (HSPs) and Mφ5 showed higher expression of microglial marker *TMEM119, GPR34*, and *CX3CR1*[35] (Fig. 4b and Supp. Data 6). In fact, Mφ1-5 showed expression of *TMEM119* while clusters Mφ6-7 and the proliferating myeloid cells did not. This is in line with recent reports highlighting two potential lineages of TAMs in glioblastoma, one that is microglia-derived and one from bone marrow-derived monocytic TAMs[21].

The largest microglia-derived TAM population is Mφ1, which was marked by higher expression of macrophage inflammatory cytokine *IL1B*, whose high expression were related to worsened survival in recurrent GBM[21]. The fraction of this cluster did not significantly change with neo-aPD1 therapy, which suggests that this population could be targeted to improve recurrent GBM patient survival. On the other hand, the proportions of the *CXCR4+* Mφ2 and ISG-high Mφ3, both of which were *MRC1*- and *MHC II+*, were significantly increased in proportion in GBM.pembro compared to GBM.rec (Fig. 4c). When we analyzed the myeloid cells from our CyTOF analysis of the expanded 34 marker set ($n = 334,530$ myeloid cells), we also observed an increased proportion of clusters c2, which is CD14+ CD11b/c+ HLA-DR+ macrophage/monocytic population (Supp. Fig. 4a–c). Of note, the ablation of *CXCR4+* myeloid population was reported to suppress melanoma and glioblastoma growth in pre-clinical models[36,37], which implies that Mφ2 is an immuno-suppressive population induced by neo-aPD1.

The observation of increased Mφ2 and ISG-high Mφ3 was confirmed by the general upregulation of genes associated with the IFN-γ pathway, such as *STAT1, IRF1, GBP4, TAP1, HLA-DQA1*, and *CXCL9/10* (Fig. 4d, e and Supp. Data 6). These changes were most pronounced in the ISG-high Mφ3 cluster. We noted that the expression of *CXCL9* and *CXCL10* has been previously found to be involved in increasing T cell trafficking

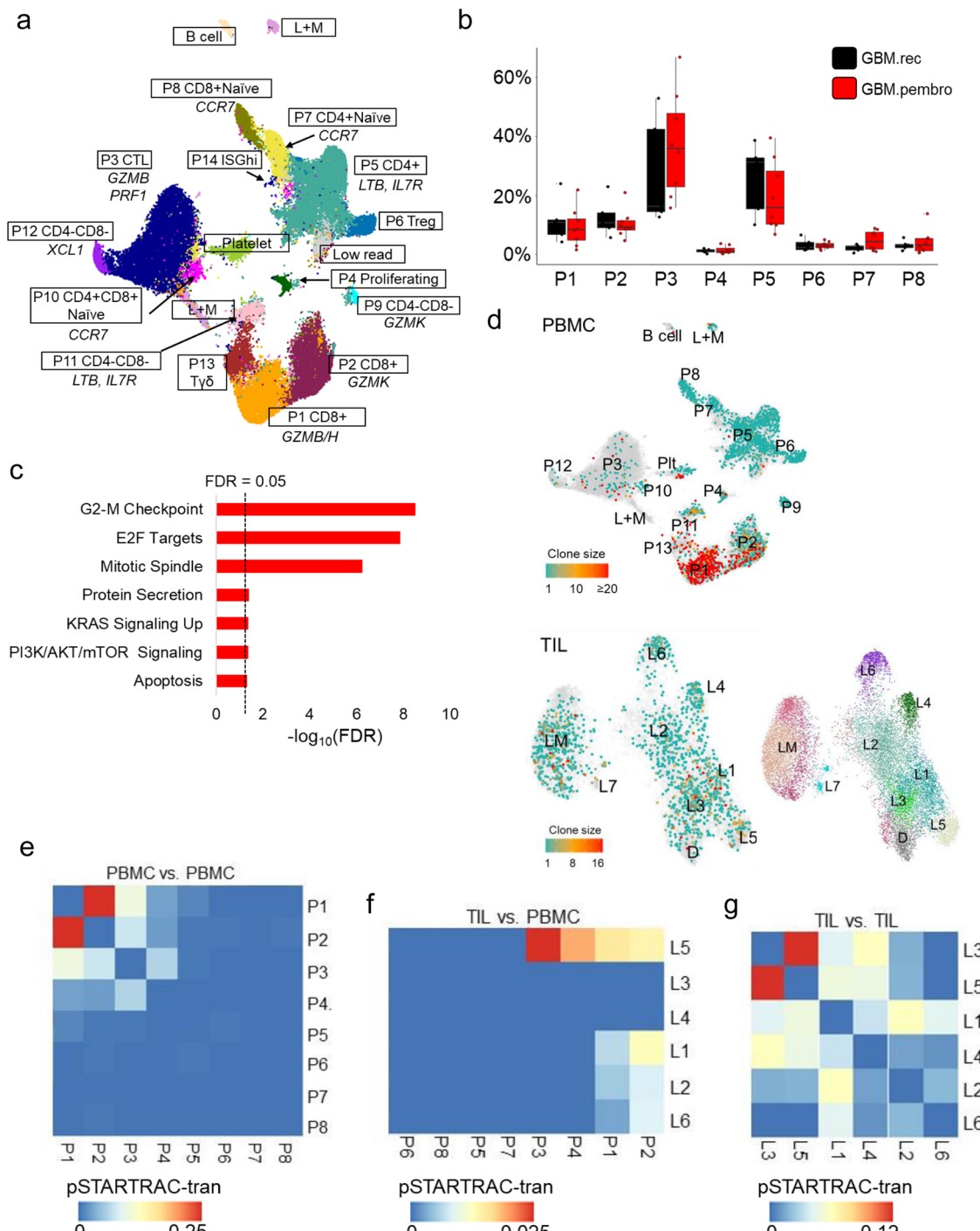

**Fig. 3 The phenotype and clonal distribution of intratumoral vs. peripheral T cells in recurrent glioblastoma patients. a** A UMAP projection of the lymphoid compartment of the peripheral blood of recurrent GBM patients analyzed using scRNAseq ($n = 56,444$ cells from 5 GBM.rec and 8 GBM.pembro patients and two healthy donor controls). **b** The proportions of each lymphoid clusters in **a** in GBM.rec (black) and GBM.pembro (red) samples. **c** MSigDB Hallmark genesets showing significant overlap with the genes upregulated in the GBM.pembro PBMC (FDR values, two-sided fisher exact test). **d** T cell clone sizes as estimated by the TCRβ clones detected in the PBMCs (top) and TILs (bottom) using TRUST4. **e**, **f**, **g** Heatmaps showing the STARTRAC analysis of the shared TCRs among the PBMC clusters (**e**), across the PBMC and TIL clusters (**f**), and among the TIL clusters (**g**). Shared clones across two clusters indicates potential transition from one cluster to the other (non-directional). The higher the fraction of the shared TCR, the more likely that the T cells in the two clusters transition from one to the other. Source data are provided as a Source Data file or in Supp. Data 4 and 5. In all boxplots, the median is indicated by the line within the box and the 25th and 75th percentiles indicated by the lower and upper bounds of the box. The upper and lower lines above and below the boxes represent the whiskers. The gene set enrichment *P* values were calculated using two-sided Fisher exact test with FDR adjustment for multiple comparisons.

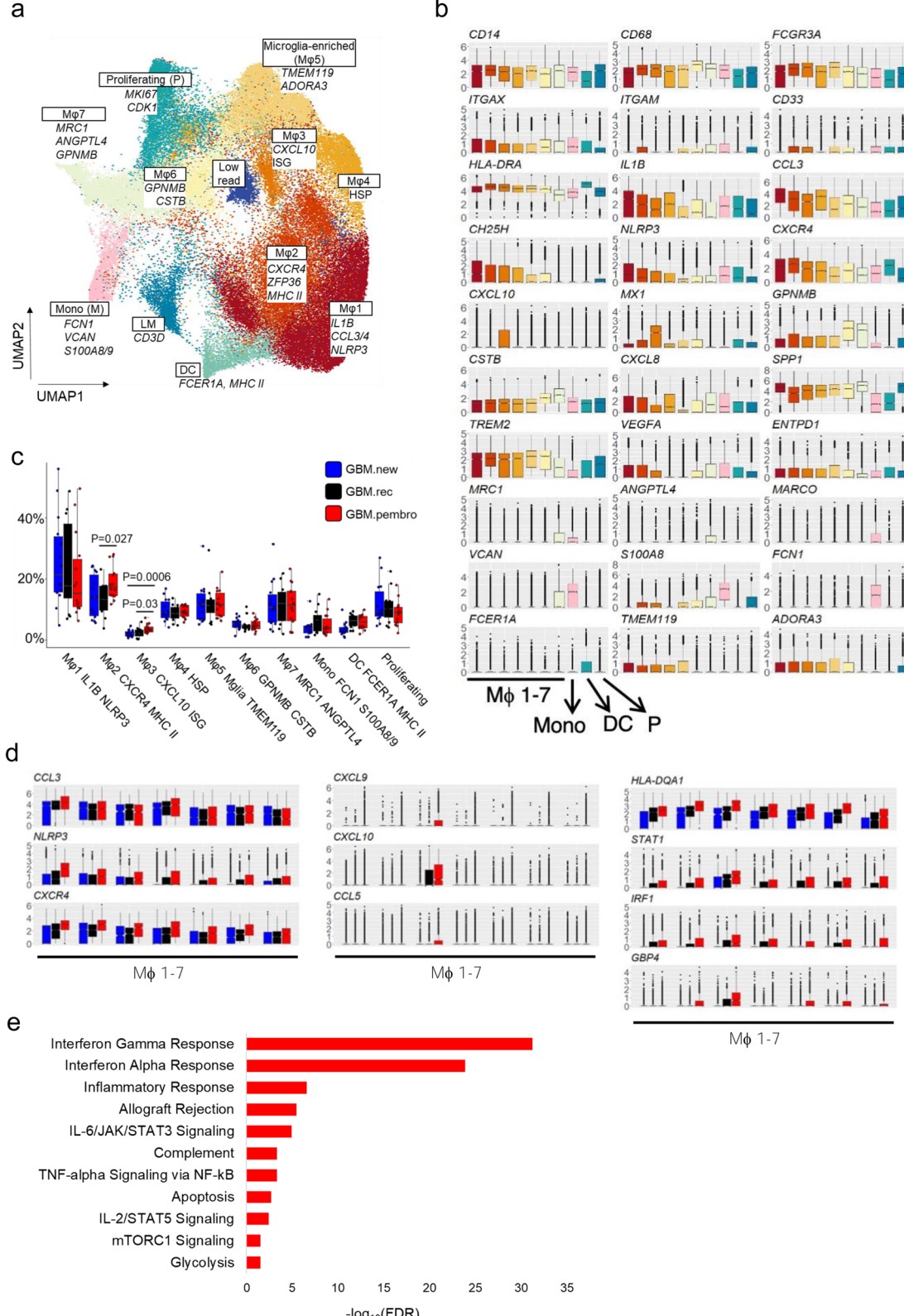

into solid tumors and involved with immunotherapy efficacy[38]. The cells in the GBM.pembro group also showed increased expression of several macrophage inflammatory genes such as: *NLRP3*, *CH25H*, *CCL3/4*, and *CXCR4*.

In the CyTOF data analysis, we found a significant increase of CD274 (PD-L1) median expression and a concomitant increase in the percentage of PD-L1+ myeloid cells in the GBM.pembro group (Supp. Fig. 4d, e and Supp. Data 7); the increased PD-L1 expression may be induced by the increased IFN-γ associated with neo-aPD1. Of note, we observed that CD274 was most significantly increased by neo-aPD1 in the CD206+ monocytic/ macrophage cluster 1, which is likely a mature macrophage population with immunosuppressive properties (Supp. Fig. 4d). The same trend of increased *CD274* transcript was observed in

**Fig. 4 Neo-aPD1 induces multiple immunosuppressive TAMs and monocyte populations. a** A UMAP projection of the myeloid compartment of the 40 patient samples analyzed using scRNAseq ($n = 72,492$ cells from 40 total patients: 14 GBM.new, 12 GBM.rec, and 14 GBM.pembro). **b** The expression levels of various marker genes of each cluster in **a. c** The proportion of the myeloid cells in each cluster across multiple tumor groups (blue: GBM.new, black: GBM.rec, red: GBM.pembro). **d** Differentially upregulated genes in GBM.pembro samples across the macrophage clusters. Colors the same as **c. e** MSigDB Hallmark genesets showing significant overlap with the union of the genes upregulated in the GBM.pembro group across all myeloid clusters (FDR values, two-sided fisher exact test). *P* values were calculated using a two-sided Wilcoxon rank-sum test unless otherwise noted. Source data are provided as a Source Data file or in Supp. Data 6. In all boxplots, the median is indicated by the line within the box and the 25th and 75th percentiles indicated by the lower and upper bounds of the box. The upper and lower lines above and below the boxes represent the whiskers. The gene set enrichment *P* values were calculated using two-sided Fisher exact test with FDR adjustment for multiple comparisons.

the scRNAseq data both in the whole myeloid fraction and in the *CD206/MRC1+* Mϕ7 cluster (Supp. Fig. 4f).

Looking at the monocyte and proliferating myeloid populations, we noted a similar increase of *STAT1, IRF1, MHC II,* and *CXCL9/10* as a result of neo-aPD1 treatment (Supp. Fig. 5a). We further analyzed the monocyte population (cluster M) by increasing the clustering resolution. Two subclusters of monocytes were identified: one classical monocyte cluster (Mono 1: *FCN1, SELL, S100A8/9*) and one non-classical monocyte (Mono 2: *LILRB2, CD48, ITGAL*)[9]. We noted a number of subclusters that nicely overlapped the macrophage populations, indicating that they arose from the differentiation of the monocytes (Supp. Fig. 5b, Supp. Data 8).

**Neo-aPD1 therapy increases the proportion of interferon activated DC populations and the proportion of CCR7+ LAMP3+ XCR1+ DCs in the tumor.** From our myeloid sub-clustering analysis, we noticed that there was a DC cluster; thus, we looked at the composition of the cDC populations in more detail. We further clustered our DC population ($n = 2,960$) from our scRNAseq analysis and generated nine subclusters (Fig. 5a, b and Supp. Data 9): 1 cDC2 cluster (cDC2: *CD1C, FCER1A, CLEC10A*), 1 cDC1 cluster (*XCR1, CLEC9A, BATF3, IRF8*), 1 activated and/or migratory DC cluster (mDC: *CCR7, LAMP3, CD80*; similar to a population recently described[21]), and 4 clusters that were monocyte-like (mo-DC1: *CD14, CD1c, FCER1A, CLEC10a*; mo-DC2: *CD14, C1QA, CD141*; mo-DC3: *CD14, C1QA, IL1B*; mo-DC4: *CD14, C1QA, CLEC10A*), 1 cluster with cells with low reads, and 1 cluster with cells that had high expression HSP transcripts.

There were no significant differences in the proportion of cells between the different conditions in any of the DC sub-clusters (Supp. Fig. 6a). However, when we examined the gene expression changes in the different DC subclusters, we noticed several significant and immunologically relevant alterations. Neo-aPD1 was associated with an overall induction of ISGs such as *STAT1, IRF1, GBP4,* and *TAP1* in multiple DC subsets (Fig. 5c). In the cDC1 population, neo-aPD1 was associated with increased expression of *XCR1.* We posit that these cDC1 cells may be attracted by the *CCL5* and *XCL1* expressing CD8+ T cells (Fig. 2a, g). In mDC cells, neo-aPD1 was associated with increased expression of the DC activation and T cell trafficking chemokine genes: *CD40, CXCL9/10,* and *IL32.* At the same time, neo-aPD1 also induced the expression of activation-induced, negative feedback DC genes such as *IDO1* and *PD-L1/L2.*

Importantly, we also observed a significantly increased DC expression of *XCL1/2, CXCL9/10, CXCR3,* and *CCL5* with neo-aPD1 (Supp. Fig. 6b). Because these genes are associated with DC and T cell trafficking, this may indicate that neo-aPD1 results in additional recruitment of T cells and DCs by the intratumoral DCs. On the other hand, the DCs in the GBM.new samples showed higher expression of monocytic genes (*CD14, CXCL8*) and GBM.rec cells showed upregulation of genes related to the type I

IFN (*IFI6, IFI27*) and NFκB (*NFKB1*) pathways (Supp. Fig. 6c), indicating a generally less activated DC population[39].

To examine the different transcriptional states in the DC populations, we further analyzed the DCs using Monocle 2. Our pseudotime trajectory analysis yielded nine distinct DC fates (Fig. 5d, e). Interestingly, we observed three separate *CCR7+* mDC populations: mDC with cDC1 characteristics (Migratory cDC1: *XCR1, CLEC9A, BATF3, XCR1, LAMP3*), one with a mixed moDC and/or cDC2 characteristics (Migratory cDC-2/moDC: *CD1C, CLEC10A, FCER1A, TREM1, S100A4/6/9*) and one *VEGFA+* cDC2-like mDC (Migratory cDC2-2: *CLEC10A, FCER1A, VEGFA, NLRP3, CD39/ENTPD1*). Notably, only the first two migratory DC populations (Migratory cDC1 and Migratory cDC-2/moDC) were induced (≥1.5-fold) by neo-aPD1 treatment. Conversely, neo-aPD1 also reduced the fractions of monocyte- and macrophage-like DC populations that were enriched in the GBM.new and GBM.rec tumors. These populations expressed high levels of known immunosuppressive myeloid markers like *MRC1, TREM2,* and *GPNMB* and T cell suppressive genes such as *VEGFA, IL10,* and *ENTPD1.*

Despite these increases in proportion of IFN-γ activated, migratory DCs with neo-aPD1, the overall proportion of the neo-aPD1 activated DCs among all cells remained relatively small (the total number of DC was 2960 out of 72,492 analyzed myeloid cells). Such limited number of activated DC likely leads to suboptimal antigen presentation within the tumor which results in inadequate antitumor activities by T cells. This implies that one potential way to achieve a longer lasting response to neo-aPD1 may be to further boost intratumoral DC activation or trafficking.

**Neo-aPD1 therapy potentiates the engagement of additional immune checkpoints.** Given the transcriptomic changes in genes encoding receptor-ligand pairs in response to neo-aPD1, we performed an unbiased cell-cell interaction inference analysis across all immune cell types using CellChat[40]. We included all annotated populations we had in the lymphoid (L1-L7), macrophage (Mϕ1-7 and P), DC (cDC1-2, mDC, moDC1-4), and the monocyte populations (Mono1-2, Mϕ-like1-4).

Immune cells from GBM.pembro patients had the greatest number of all possible inferred interactions. Both GBM.rec and GBM.pembro showed a similar average interaction strength, which assesses the probability of observing an interaction given the expression levels of the receptor and its ligand (Fig. 6a). Next, we calculated the signaling pathways with the highest combined interaction strength ratio between GBM.pembro and GBM.rec. The pathways enriched in GBM.pembro included: TIGIT:CD226: NECTIN2, CD86:CTLA4, PDCD1:PD-L1/L2, Type II IFN (IFN-γ), CXCL:CXCR, XCR:XCL, and CCR:CCL (Fig. 6b). Notably, these were genes that previously showed relative increases in transcriptional levels following neo-aPD1. On the other hand, neo-aPD1 decreased the signaling pathways VEGF and ANGPTL, which are known wound healing and angiogenic processes correlated with worse survival in GBM[41].

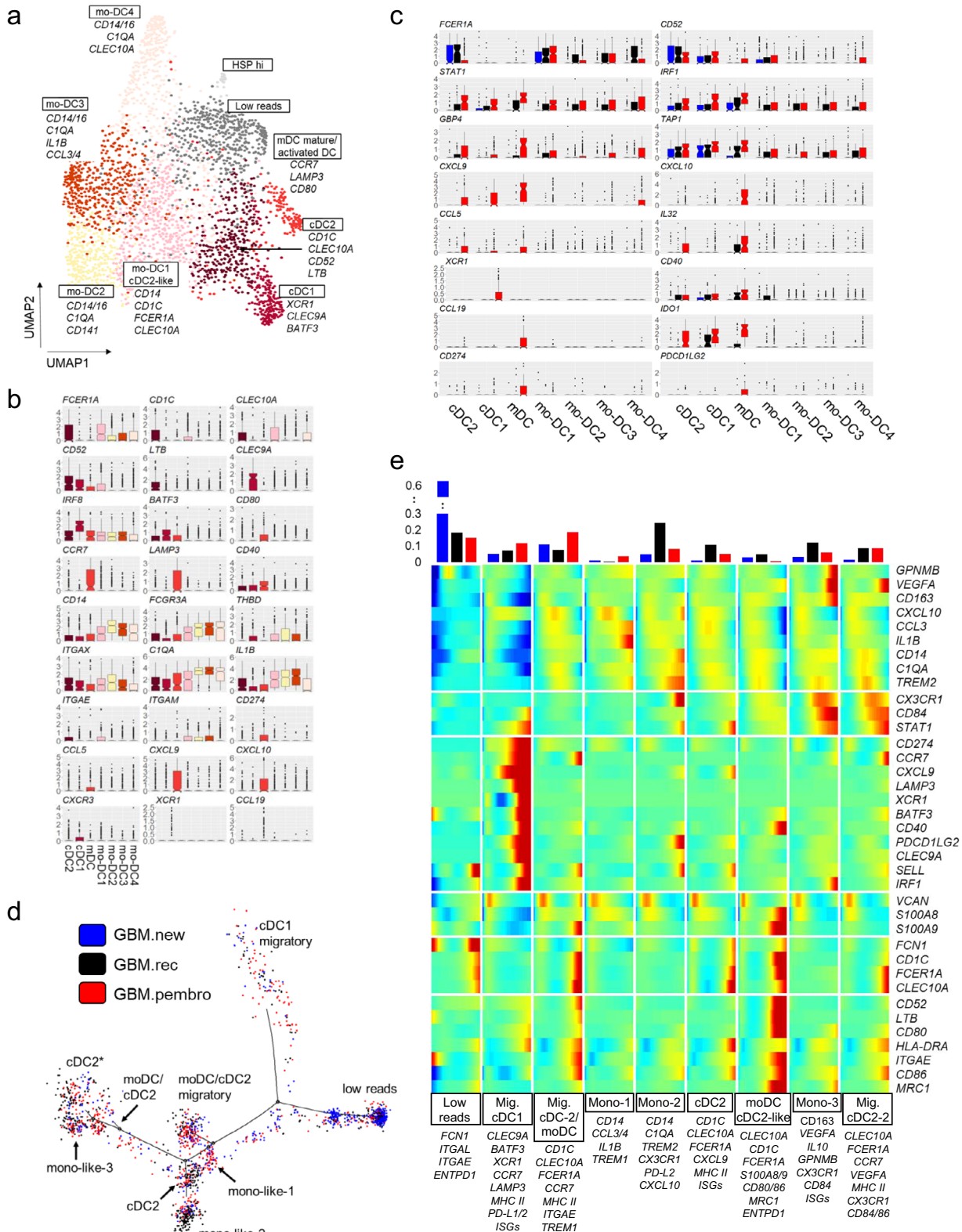

**Fig. 5 Neo-aPD1 therapy increases the proportion of intratumoral migratory DCs. a** A UMAP projection of the DC compartment analyzed using scRNAseq (n = 2960 cells from 40 total patients: 14 GBM.new, 12 GBM.rec, and 14 GBM.pembro). **b** The normalized expression of the marker genes of different DC clusters. **c** Differentially upregulated genes in GBM.pembro samples in one or more DC clusters (blue: GBM.new, black: GBM.rec, and red: GBM.pembro). **d** Pseudotime projection of all DCs as analyzed by Monocle 2. Colors are as in **c**. **e** Heatmap showing the expression of DC/monocyte marker genes in each DC fate as defined in **d**. (top) The proportion of DCs from each tumor groups in each fate. Source data are provided in Supp. Data 9. In all boxplots, the median is indicated by the line within the box and the 25th and 75th percentiles indicated by the lower and upper bounds of the box. The upper and lower lines above and below the boxes represent the whiskers.

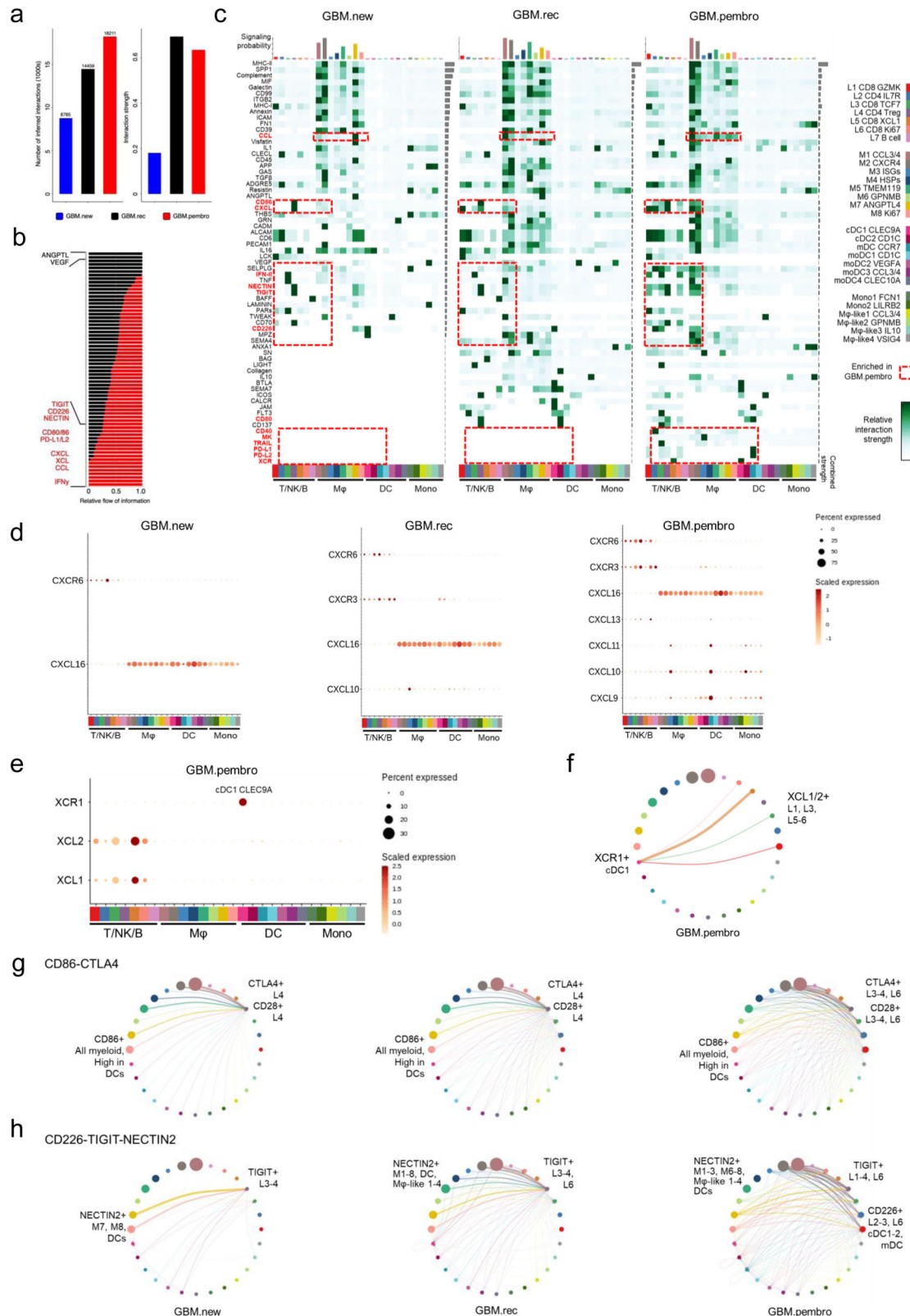

We next examined the signaling probability of all receptor-ligand interactions found in our previously identified cell clusters (Fig. 6c). The majority of cellular interactions were dominated by the macrophage populations and tended to remain stable across GBM.new, GBM.rec, and GBM.pembro patient samples. Of note, the number of possible pathway interactions involving the T/NK clusters were higher in GBM.pembro compared to GBM.rec. We

also noted increased interaction strength between *CXCL9/10/11* and *CXCR3*+ T/NK cells in the GBM.pembro compared to only *CXCL10-CXCR3* in GBM.rec (Fig. 6d). Furthermore, neo-aPD1 induced the recruitment of *CCR5*+ T cells potentially by the overexpression of *CCL3/4* (Mφ1-4) and/or *CCL5* (L1, 3, 5) (Supp. Fig. 7a). *CCR5* expression by both CD4+ and CD8+ T cells was reported to be crucial in boosting anti-tumor responses by

**Fig. 6 Increased engagement of additional immune checkpoints may limit neoadjuvant anti-PD1-driven T cell activation. a** The total absolute (left) or normalized (right) number of CellChat-inferred interactions among the population in each tumor group (blue: GBM.new, black: GBM.rec, red: GBM.pembro, n = 156,766 cells from 40 total patients: 14 GBM.new, 12 GBM.rec, and 14 GBM.pembro). **b** Signaling pathways with significant differences in the overall information flow (sum of all normalized interaction) between GBM.rec and GBM.pembro. **c** The overall signaling patterns between CellChat curated pathways and our defined cell clusters. The interaction strength reflects the sum of all normalized interaction in each pathway. The top bar graph compares the sum of total interaction strength per cell type while the bar graph on the right summarize the interaction strength per pathway. Of note, myeloid cell interactions dominate the bulk of the interaction across all tumor groups. **d**, **e** The normalized expression of CXCL (**d**) and XCL pathway genes with significant expression in indicated tumor groups. The genes shown are known ligand receptor pairs (CXCL16-CXCR6, CXCL9/10/11-CXCR3, and XCL1/2-XCR1). **f** The inferred XCL1/2-XCR1 signaling network among the cell populations represented by the nodes. Color code as in **e**, size of node represents the size of the population, edge width represents the pathway specific interaction strength. **g**, **h** Same as **f** for CD86-CTLA4-CD28 signaling (**g**) or CD226-TIGIT-NECTIN (**h**) pathways. Both T cell checkpoints were inferred to be significantly engaged after neo-aPD1. Source data are provided as a Source Data file.

optimizing helper-dependent CD8+ T cell priming at immunological synapse[42,43]. CellChat also highlighted GBM.pembro specific chemokine receptor-ligand interactions between *XCR1* (cDC1) and *XCL1/2* (mainly expressed by the cytotoxic L5) which were not observed at all by immune cells from the other groups (Fig. 6e, f). The interaction hints at the recruitment of cDC1 by the intratumoral cytotoxic T cells.

We noted the possible engagement of two different immune checkpoint pathways: the CTLA-4 and TIGIT pathways. Specifically, GBM.pembro samples showed increased potential interactions between *CD86/CD80* and *CTLA4* and *CD28*. *CD86* was expressed by all myeloid cell clusters where the larger clusters contributed to higher interaction strength (thicker lines, Fig. 6g). At the same time, the other binding partner of *CD28* and *CTLA4*, *CD80*, was also expressed higher in recurrent GBM samples regardless of neo-aPD1 (Supp. Fig. 7b). In GBM.new and GBM.rec, myeloid-expressed *CD86* and *CD80* were expected to mostly interact with the constitutively expressed *CTLA4* on T$_{reg}$ (L4) (Fig. 6g and Supp. Fig. 7b). However, in GBM.pembro, both *CTLA4* and *CD28* were induced also on the progenitor exhausted-enriched CD8+ T cell cluster (L3) and proliferating T cell cluster (L6) (Supp. Fig. 7b). The new induction of *CTLA4* in GBM.pembro likely suppresses T cell activity, implying that the potential use of anti-CTLA-4 antibody blockade may enhance the co-stimulatory signaling through *CD80/86*'s engagement of T cells' *CD28*[44] in GBM.pembro patients.

Another immune checkpoint interaction that was higher in the GBM.pembro group was between the ligands *CD226* and *TIGIT* and their receptor *NECTIN2*[45,46] (Fig. 6h and Supp. Fig. 7c). In both GBM.rec and GBM.pembro, *CD226* and *TIGIT* also interacted with the poliovirus receptor (PVR) (Supp. Fig. 7c). Just like *CTLA4* and *CD28*, *TIGIT* and *CD226* are T cell co-inhibitory and co-stimulatory receptors, respectively[47,48]. In all three conditions, interactions between *TIGIT* and *NECTIN2* remained intact, but GBM.pembro showed new interactions involving *TIGIT*+ L1 and L2 clusters. Moreover, neo-aPD1 significantly increased interactions between *Nectin2*+ myeloid clusters with *CD226*+ L2, L3, and L6 lymphoid clusters whereas this interaction was absent in GBM.new and GBM.rec (Fig. 6h and Supp. Fig. 7c). Importantly, the presence of *CD226*+ CD8+ T cells has been shown to be a pre-requisite for response to anti-TIGIT immunotherapy[49].

Together, the interactome analysis shows that neoadjuvant anti-PD-1 therapy increases the number of novel receptor-ligand pairs that represent potential therapeutic targets to inhibit or augment in addition to the PD-1/PD-L1 axis.

## Discussion

In this study, we utilized high-dimensional single-cell analyses to understand the characteristics of tumor-associated myeloid cells and lymphoid cells in glioblastoma patients treated with or without neoadjuvant anti-PD-1 checkpoint blockade immunotherapy. We

found that the tumor microenvironment was heavily dominated by myeloid cells, but in patients treated with neo-aPD1, there was an increase in the density of T cells. Single-cell RNAseq analysis showed that neo-aPD1 increased the proportion of cytotoxic T cells (P3) in peripheral blood, implying systemic T cell activation. Single-cell TCR analysis of the PBMC and TILs showed TCR overlap between tumors' and PBMC's activated/cytotoxic T cell clusters (L1 and P1-2, L5 and P1-4), which suggests that these T cells migrate from the systemic circulation to the tumor microenvironment and may underlie clonal replacement observed after neo-aPD1[50,51]. Moreover, within the tumor, we observed TCR overlap between cytotoxic T cells (L5) and a population of progenitor exhausted T cells (L3). This indicates a transition of cytotoxic effector T cells into progenitor exhausted T cells, resulting in a fractional increase of the L3/Exh-progenitor population. We are aware of the inherent limitations in TRUST4's sensitivity for TCR detection to our 3′, poly-A enrichment-based RNA-seq analysis. It is likely that most of the TCR clones we detected would be from larger TCR clones. Nevertheless, the overlaps of these major clones can still suggest the direction of phenotypic transition across different T cell clusters in the periphery and TILs. Many of these tumor-infiltrating T cells highly expressed chemotactic genes, such as *CCL5* and *XCL1*, that can attract additional T cells and professional antigen presenting cells (e.g., dendritic cells) into the tumor microenvironment.

In our dendritic cell analysis, we found that neo-aPD1 was associated with increases in expression of genes related to DC trafficking and also proportions of IFN-γ activated, cross-presenting DC subsets (DC pseudotime Migratory cDC1 and Migratory cDC-2/moDC). Our interactome results also indicated that chemotactic factors produced by intratumoral T cells may be responsible for DC trafficking along the *XCL1/XCR1* axis. However, despite the increase in proportion of these migratory DC subsets, these cells remained a relatively small percentage of the whole APC populations. As such, the use of CD40 agonists to expand these activated, cross-presenting DC populations may be warranted as it was reported to result in more effective T cell priming by the DCs in a murine colorectal cancer setting[38].

In the TAM and monocyte populations, neo-aPD1 was associated with elevated expression of genes related to IFN-γ stimulation, including those related to T cell trafficking, such as *CXCL9/10*. Importantly, the receptor for *CXCL9/10*, *CXCR3*, is expressed by the multiple CD8+ T cell populations and our interactome analysis confirmed likely increased interactions along this axis. Importantly, neo-aPD1 also specifically increased potential interactions along the immunosuppressive *CTLA4:CD86/CD80* and *TIGIT:Nectin2/PVR* axes between myeloid and the activated T cell populations (L1, L3, and proliferating L6). The upregulation of *CTLA4* and *TIGIT* were both accompanied by the upregulation of their respective co-stimulatory molecules *CD28* and *CD226*, which strongly suggests that these T cells are primed to be activated further by blocking CTLA-4[52] and/or TIGIT[45,48]. In fact, CTLA-4 blockade was

shown to elicit a stronger intracranial response in melanoma brain metastasis[53], suggesting the feasibility of testing this combination in a neoadjuvant setting for recurrent GBM. As such, we have recently started a Phase I clinical trial examining the dual treatment of neoadjuvant CTLA-4 and anti-PD-1 in the recurrent GBM setting (NCT04606316). Additionally, targeting the other neo-aPD1-induced immune checkpoint molecule, TIGIT, could also be beneficial to further boost the intratumoral T cell activity.

Putting all of our findings together, we propose the following model for how neo-aPD1 affects the immune landscape of recurrent GBM (Supp. Fig. 8). (1) Neo-aPD1 systemically activates T cells in the periphery (P1-4) and some of these T cells migrate into the GBM tumor microenvironment. (2) The activated T cells (L1, L5) produce cytotoxic granules like Granzyme B and K and perforin, which promote tumor death, chemotactic factors like CCL5 and XCL1/2, which signal to cDCs and/or T cells in the blood to enter the tumor microenvironment, and IFN-γ, which activates these incoming cDCs. (3) The IFN-γ activated cDCs produce CXCL9/10/11 to promote additional DC and T cell trafficking into the tumor and then cross-present and activate T cells. Increased T cell activation produces more IFN-γ in the microenvironment that results in a positive loop that promotes additional immune cell infiltration. In the process, a population of tumor antigen-specific CD8+ T cells started to transition into a progenitor exhausted phenotype (L3, CD8+ pseudotime, Exh-progenitor). (4) The increased levels of IFN-γ also recircuits the more numerous myeloid populations resulting in some myeloid populations that produce factors, like CXCL9/10, that increase intratumoral T cell trafficking (Mφ3). However, the majority of the myeloid populations produce IFN-γ stimulated inflammatory molecules such as IL-1ß (Mφ1), express immunosuppressive surface markers like CXCR4 (Mφ2) and PD-L1 (Mφ7). Importantly, all of the TAM and monocyte populations maintained a high expression of CD86 and NECTIN2, which potentially engage neo-aPD1-induced CTLA-4 and TIGIT T cell checkpoints on the activated T cell populations. This enhanced inhibitory interaction may drive the increase of the progenitor exhausted T cells and limit the magnitude and persistence of antitumor T cell activities.

In this paper, we set out to characterize tumor-infiltrating immune populations in patients with glioblastoma. Our integrated analyses have detailed the immune landscape in these malignant brain tumors at the single-cell level and how this immune landscape changes with neo-aPD1 therapy. As our study did not include comparison of patient-matched pre- and post-treatment samples, we are unable to conclusively state that the anti-tumor immune effects is a direct result of neoadjuvant anti-PD-1 in each patient. However, we evaluated equally sized groups of patients at a single timepoint, which can shed light into the early effects of PD-1 blockade on the systemic and intratumoral microenvironment. Future work studying the timing of neoadjuvant anti-PD-1 administration both in the pre-clinical and clinical settings would be needed to confirm these findings. Nevertheless, for now, our data would suggest that although neo-aPD1 improves survival outcomes by increasing anti-tumor T cell responses, it is not completely curative as this T cell response is curtailed by the engagement of additional T cell checkpoints and other immunosuppressive pathways by the myeloid populations. We postulate that further efforts to increase effector T cell and DC recruitment, in combination with blocking the immunosuppressive signaling engaged by the myeloid cells, will be necessary to achieve clinically relevant effects in recurrent glioblastoma. As neoadjuvant studies progress, we hope that others will be able to contribute to this trove of information to hone our understanding of this uniformly fatal tumor.

## Methods

**Ethical compliance**. Research conducted in this study complied with all relevant ethical regulations.

The UCLA Medical Institutional Review Board 2 (IRB#10-000655-AM-00059) approved all protocols related to patient specimen collection.

**Patient tissue collection and single-cell isolation**. Tumor tissue was obtained from tumors of patients who underwent surgery at the University of California, Los Angeles. All patients provided written informed consent. This study was conducted in accordance with the Declaration of Helsinki and under an institutional review board approved protocol. All GBM patients (newly diagnosed and recurrent) were treated with standard of care therapies. Patients with surgically accessible recurrent GBMs, regardless of recurrence number and who did not show response to standard of care treatment and had tumor progression, were treated with off-label, off-trial pembrolizumab 200 mg by intravenous infusion 14 ± 5 days before surgery, similar to our previously published study[4]. Peripheral blood was drawn from patients prior to surgical resection of their tumors. Tumor tissue not needed for diagnosis was digested using the Miltenyi Brain Tumor Dissociation kit (Miltenyi Biotec, cat. 130-095-42) and gentleMACS dissociator (Miltenyi Biotec, cat. 130-093-235) and labeled with CD45+ microbeads (Miltenyi Biotec, cat. 130-045-801). CD45+ cells were positively selected for with Miltenyi LS columns (Miltenyi Biotec, cat. 130-042-401) and MidiMACS separator (Miltenyi Biotec, cat. 130-042-302). Collected CD45+ cells were then placed in Bambanker (Fisher Scientific, cat. 302-14681) and stored in liquid nitrogen. Peripheral blood mononuclear cells were collected in CPT tubes (BD Biosciences, cat: 362753), isolated according to the manufacturer's protocol, placed in freezing media made of 90% human AB serum (Fisher Scientific, cat. MT35060CI) and 10% dimethyl sulfoxide (Sigma, cat. C6295-50ML) and stored in liquid nitrogen. Most of the patient samples used in this study were exhausted to gather sufficient amount of data for analysis. Any remaining sample will be available from the corresponding author upon reasonable request, though the quantity may or may not be sufficient to complete all the corresponding experiments.

**Time-of-flight CyTOF mass cytometry**. Peripheral blood mononuclear cells and tumor-associated CD45+ cells were collected at the time of surgery as described above. On the day of data acquisition, samples were briefly thawed in a 37 °C water bath and washed in RPMI-1640 media (Gennesse Scientific, cat: 25-506) supplemented with FBS and penicillin and streptomycin. Cells were then prepared for mass cytometry analysis according to the Maxpar cell surface staining protocol. Briefly, 0.5–3 × 10⁶ cells were washed with PBS and treated with 0.1 mg/mL of DNAse-1 Solution (StemCell Technologies, cat: 07900) for 15 minutes at room temperature. Cells were then resuspended in 5 μM Cell-ID cisplatin (Fluidigm, cat: 201064) as a live/dead marker for 5 min at room temperature. After quenching with the Maxpar cell staining buffer (Fluidigm, cat: 201068), the cells were incubated with a 24- to 34-marker panel for 30 min at room temperature (Supp. Data 1, all antibodies were stained at a 1:50 dilution). After washing with cell staining buffer, cells were incubated overnight in 125 nM iridium intercalation solution (1000X dilution of 125 μM Cell-ID Intercalator-Ir; Fluidigm, cat: 201192A) in Maxpar Fix and Perm Buffer (Fluidigm, cat: 201067) to label intracellular DNA. Cells were then washed with cell staining buffer and distilled water. Due to likely enzymatic degradation of the CD8 co-receptor (the same CD8 antibody worked on control PBMCs, data not shown), we only performed general CD3+ T cell analysis. CD68 also showed significant batch effects (data not shown) that could not be corrected and was removed from downstream analyses. Finally, we also noted that the magnitude of PD-1 (CD279) protein expression in GBM.pembro patients was specifically lower than GBM.new and GBM.rec, which likely indicates competition between the CyTOF antibody targeting CD279 and the humanized pembrolizumab antibody (data not shown).

Events were subsequently acquired on a Helios mass cytometer (Fluidigm) in the University of California, Los Angeles Jonsson Comprehensive Cancer Center Flow Cytometry core. After acquisition, all fcs files were normalized together using the R package *premessa* version 0.2.4 with the four element calibration beads (Fluidigm, cat: 201078). After normalization, live singlets were gated. Each markers' intensities were capped at 1st and 99th percentile, normalized from 0 to 1 and centered at the mean. Up to 20,000 cells were randomly subsampled from each sample. Dimensional reduction was performed using the Python implementation of UMAP using the *reticulate* R package version 1.18. Unsupervised clustering was carried out by *PhenoGraph* or *ClusterX* algorithm using R package *cytofkit*[54] version 1.4.10. The median expression of each marker in each cluster was visualized by R package *pheatmap* version 1.0.12 and immune cell populations were identified based on the expression of specific markers.

**Multiplex fluorescent immunohistochemistry**. Formalin-fixed paraffin-embedded tissue sections of six recurrent GBM patients were stained for multiplex immunohistochemistry (mIF) to spatially visualize and quantify the tumor microenvironment. The following marker panel was used for staining by the Translational Pathology Core Laboratory (UCLA) using the Bond RX Fully Automated Research Stainer (Leica Biosystems, cat: 21.2821) and Opal Polaris 7 Color Automation IHC Detection Kit protocol (Leica Biosystems, cat:

NEL871001KTIHC): CD14, CD206, HLA-DR, CD4, CD45, and CD8. Prior to antibody application, slides underwent a thirty-minute epitope retrieval process for CD14, HLA-DR, CD4, and CD8 using pH9 BOND epitope retrieval solution 2 (Leica Biosystems, cat: ER2). CD206 and CD45 used a pH6 BOND epitope retrieval solution 1 (Leica Biosystems, cat: ER1). Antibody clones and dilutions were performed in the following order conjugated to an Opal tyramide signal amplification reagent: CD14 (1:3500, Abcam, clone: EPR3653, cat: ab133335) with Opal 480, CD206 (1:300, Sigma-Aldrich, clone: CL0387, cat: AMAB90746-25UL) with Opal 520, HLA-DR (1:300, Abcam, clone: EPR3692, cat: ab92511) with Opal 570, CD4 (1:50, Dako, clone: 4B12, cat: M731029-2) with Opal 620, CD45 (1:200, Dako, clone: 2B11 + PD7/26, cat: IR75161-2) with Opal 690, and CD8 (1:200, Dako, clone: C8/144B, cat: IR62361-2) with Opal 780. All slides were counter-stained with spectral DAPI (Akoya Biosciences) for nuclear detection. Multispectral whole tissue imaging of all slides was performed at a 40x magnification (0.25 µm/pixel) using the Vectra Polaris Automated Quantitative Pathology Imaging System (Akoya Biosciences, cat: CLS143455) by UCLA's Translational Pathology Core Laboratory. Whole tissue scanned images were then exported as a MOTiF Digital Slide format (.qptiff) file to allow for further viewing and ROI selection on Phenochart v1.0.12 (Akoya Biosciences). All images were spectrally unmixed using inForm image analysis software to identify and separate overlapping background autofluorescence per Opal and to provide a more accurate in situ visual for quantification.

All slides were spatially examined and quantified using HALO Image Analysis Software v3.0.311.398 (Indica Labs). Multilayered TIFF files from Inform were stitched together using HALO to reconstruct the unmixed whole tissue image. Appropriate nuclear segmentation and positive dye threshold intensities were determined for an overall whole tissue marker analysis. Tissue regions of interest were selected while autofluorescent, non-specific background was excluded. Multiplex IF raw data will be freely available to interested researchers.

**Single-cell RNA sequencing**. CD45+ cells were isolated using a Miltenyi bead pulldown and immediately frozen for batched analysis, as described above. Cell preparation, library preparation, and sequencing were carried out according to Chromium product-based manufacturer protocols (10X Genomics). Sequencing was carried out on a Novaseq6000 S2 2x50bp flow cell (Illumina) utilizing the Chromium single-cell 3′ gene expression library preparation (10X Genomics), per manufacturer's protocol at the Technology Center for Genomics and Bioinformatics Core, UCLA. Data were demultiplexed and aligned with Cell Ranger version 3.0.0 or higher (10X Genomics) and aligned to the Genome Reference Consortium Human Build 38 (GRCh38). Data were then imported and analyzed with the Seurat package for R[55] version 3.1.5. For quality assurance, we examined the number of features per cell and percent mitochondrial RNA in each sample. Cells with >20% mitochondrial features were excluded from further analysis. The raw transcript count for each sample was individually normalized using the *NormalizeData* function. The Seurat data object from each sample were then integrated into one large Seurat object. Integrated expression values were further scaled by regressing out the percent mitochondrial features, cell cycle score[56], and the number of detected genes. For dimension reduction, we ran principal component analysis, uniform manifold approximation and projection (UMAP). Different cell cluster populations were defined using the *FindNeighbors* function and the genes that were differentially expressed in each cluster or treatment was computed using the *FindMarkers* or *FindAllMarkers* function.

**Single-cell pseudotime trajectories**. From all CD45+ cells analyzed by Seurat, we selected T cell and dendritic cell populations to separately do pseudotime trajectory analysis using Monocle 2.12.0[27]. First, we excluded lymphoid-myeloid doublets from this analysis. In particular, from the T cell clusters identified by Seurat, we removed any cells with expression of *CD14* or *CD68*. Secondly, we separated CD4 and CD8 T cells by choosing cells with either at least one CD4 transcript and no CD8 transcript or the other way around. DC cells were selected from the Seurat's DC cluster, from which we removed cells with any expression of *CD3D/E/G* or *CD8A/B*. After selecting the "pure" CD4, CD8 T cell, and dendritic cell populations, we extracted the normalized gene-cell expression matrix of each population to construct the CDS object in Monocle2. We also used Seurat's *FindVariableFeatures* function to choose the top 1000 most variable genes as the ordering features for trajectory construction. Dimension reduction was performed using *Discriminative Dimensionality Reduction with Trees (DDRTree)* algorithm. The pseudotime trajectory was then learned by *orderCell* function in Monocle2. The expression of selected genes was visualized using Monocle2's plot *multiple_branches_heatmap* function.

**T cell receptor analysis**. We inferred CDR3 sequences of T cell receptors from the single-cell RNAseq bam files using the *TRUST4* software version 1.0.2. Cells with at least one productive TCRβ chain were kept for subsequent analysis. Cells with the same TCRβ sequence were considered to be one TCR clone. A clone with at least two cells in a given population was defined as an expanded clone. The likelihood of two cell populations to share clones was defined by the transition index computed using *TCR TRACking (STARTRAC)* method version 0.1.0. We noted that the

number of TCR clone detected per patient is generally low and, as such, we were only able to perform a group-based analysis to describe the overall transition patterns of T cells from different TIL and PBMC clusters.

**Interactome analysis**. We analyzed our scRNAseq data for potential ligand receptor (LR) interaction using the CellChat R package version 1.1.1[40]. Specifically, we analyzed potential interactions among the cells from the 28 immune subpopulations that we defined in our lymphoid (L1-L7), macrophage (Mφ1-8), DC (cDC1-2, migratory DC/mDC, moDC1-4), and monocytes (classical monocyte 1-2, Mφ-like1-4); cells from the tumor-normal like clusters, doublets, and unknown clusters were excluded from this analysis. We first generated separate cellchat objects for the GBM.new, GBM.rec and GBM.pembro cells (following closely the analysis steps for single dataset in https://github.com/sqjin/CellChat). Subsequently, we proceeded with comparison analysis in order to infer differentially enriched ligand receptor interaction among the tumor groups. The interaction comparisons were performed using the *compareInteractions* and *RankNet* functions. We visualized the significant pathway LR interactions across all tumor groups using the *netAnalysis_signalingRole_heatmap* function, setting the option *pattern = "all"* to aggregate the incoming and outcoming signaling strengths. Dot plots showing the normalized expression levels (color) and fraction cell expressing the genes (size of dot) in specific pathways were visualized using the *plotGeneExpression* function with the option *type = "dot"*. The circle plots depicting interactions among different immune populations in each tumor group were done by applying *netVisual_aggregate* with the option *layout = "circle"*.

**Statistics and reproducibility**. The statistical tests used to calculate the P-values are indicated in respective figures. Other than the differentially expressed genes called by Seurat, all P-values calculated for the boxplots and violin plots were nominal (unadjusted) P-values. Statistical analysis of the data presented in the violin plots was done using GraphPad Prism version 8.4.0.

**Reporting summary**. Further information on research design is available in the Nature Research Reporting Summary linked to this article.

## Data availability

Single-cell RNA sequencing raw data files are available in the Gene Expression Omnibus under accession number GSE154795 and CyTOF raw data files are stored at the Flow Repository under the ID FR-FCM-Z4LX. These files supply the data for Figs. 1–6 and Supplementary figs. 1–7. The raw single-cell RNA sequencing was aligned to the Genome Reference Consortium Human Build 38 (GRCh38, GCA_000001405.15) prior to analysis. The remainder of data that support the findings of this study, including multiplex IHC images and data files and processed CyTOF fcs files ready for analysis, are available from the corresponding author upon reasonable request. Source data are provided with this paper.

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

## Acknowledgements

This study was funded in part by the National Institutes of Health SPORE in Brain Cancer (P50CA211015) and NIH/NCI grant (1R01CA222695-01), NIH National Center for Advancing Translational Science - UCLA CTSI (UL1TR001881), the Parker Institute for Cancer Immunotherapy, the Brain Tumor Funder's Collaborative, and the Cancer Research Institute. Mass cytometry was performed in the UCLA Jonsson Comprehensive Cancer Center (JCCC) Flow Cytometry Core Facility that is supported by NIH award P30 CA016042. The purchase of the Helios/CyTOF mass cytometer that was used in this work was, in part, supported by funds provided by the James B. Pendleton Charitable Trust. Single-cell RNA sequencing was performed by the UCLA Jonsson Comprehensive Cancer Center Genomic Shared Resource. A.H.L. is a pre-doctoral fellow supported by the UCLA Tumor Immunology Training Grant (USHHS Ruth L. Kirschstein Institutional National Research Service Award # T32 CA009120). L.S. was supported by a Career Enhancement Program award from the UCLA SPORE in Brain Cancer. J.C.K. received a grant from the Swiss Cancer Research foundation (project BIL KFS-4563-08-2018) and the Kurt and Senta Herrmann foundation. W.H. was supported by grants from the NIH/NCI (1R01CA236910), the Melanoma Research Alliance (https://doi.org/10.48050/pc.gr.75700), the Margaret E. Early Medical Research Trust Grant, and the Parker Institute for Cancer Immunotherapy at UCLA. The schematic was created with BioRender.com

## Author contributions

R.M.P., A.H.L., L.S., A.Y.M., J.R., and W.H. designed experiments. A.H.L., L.S., A.Y.M., J.O., F.C., and J.R. performed the experiments unless specified. A.H.L., L.S., A.Y.M., J.C.K., and W.H. participated in the processing and analysis of CyTOF and scRNAseq data. R.M.P., F.C., A.Y.M A.H.L., L.S., W.H., D.N., T.C., R.G.E. and L.M.L. participated in the design, preparation, and analysis of experiments and acquired and analyzed data related to human samples. A.H.L., L.S., A.Y.M., F.C., W.H. and R.M.P wrote and revised the manuscript. All authors edited and reviewed the manuscript.

## Competing interests

R.M.P. and T.C. were paid consultants for the Merck Advisory Board. All remaining authors declare no competing interests.
