## [Peer Review File · Nature Communications]

Reviewers' Comments:

Reviewer #1:

Remarks to the Author:

The study of Lee et al focuses on high resolution analysis of the lymphoid and myeloid cells within GBM. In total, there are 70 patients analyzed—28 newly diagnosed GBM, 22 with recurrent GBM that have not seen immunotherapy, and 20 with recurrent GBM that have been treated neoadjuvantly with anti-PD-1. Immune cells were characterized by mass cytometry and by single cell RNA sequencing, with many samples characterized by both. This study includes the most comprehensive analysis by single cell methodologies of the immune cell compartment in GBM to date.

This manuscript has been through iterative reviews at this point and merits publication with only minor revisions. There are no seismic discoveries in this study, but the sheer amount of characterization data is very impressive and will be of great interest to many.

I have a few small points to be addressed:

1. could the authors discuss more about the relatively small percentage of clonally expanded cells in the TIL (Fig 3b)?
2. I believe Figure 6 and 7 are extraneous. Figure 6 would be appropriate to include, perhaps largely as supplemental, but Figure 7 is more appropriate for a Review and should be excluded here.

Reviewer #2:

Remarks to the Author:

The authors have performed adequate revisions.

Reviewer #3:

Remarks to the Author:

This has been the third time I have reviewed this manuscript and I have seen the progression of this work to a more elegant and nuanced discussion of how neoadjuvant pembro treatment alters the GBM microenvironment.

The authors responded to all of my queries from the previous two submissions, and because of this, I don't have much more to add in terms of reviewer concerns. I like The addition of potential interactions (CellChat) and implications for future therapies (anti-TIGIT). I am also happy with the de-emphasis on the T-cell cytof data as that was my biggest concern.

Below is a minor concern I have regarding the pseudo time trajectory analysis of DCs. The authors suggest that there are changes in DC subsets in Figure 5. however, in the pseudotime analysis, it appears the majority of DC in the newly diagnosed GBM are "low reads". Could this preponderance of low-read cells influence your analysis? Is it possible you see a more inflammatory DC population in neoPD1 because of the differential quality in the DC data?

REVIEWERS' COMMENTS

Reviewer #1 (Remarks to the Author):

The study of Lee et al focuses on high resolution analysis of the lymphoid and myeloid cells within GBM. In total, there are 70 patients analyzed—28 newly diagnosed GBM, 22 with recurrent GBM that have not seen immunotherapy, and 20 with recurrent GBM that have been treated neoadjuvantly with anti-PD-1. Immune cells were characterized by mass cytometry and by single cell RNA sequencing, with many samples characterized by both. This study includes the most comprehensive analysis by single cell methodologies of the immune cell compartment in GBM to date.

This manuscript has been through iterative reviews at this point and merits publication with only minor revisions. There are no seismic discoveries in this study, but the sheer amount of characterization data is very impressive and will be of great interest to many.

I have a few small points to be addressed:

1. could the authors discuss more about the relatively small percentage of clonally expanded cells in the TIL (Fig 3b)?

- The author brought up an important point. We noted that most of the large clones in the PBMC actually were contributed by two samples: LB4143P and LB4130P (Suppl. Table 5); we are not sure of the reason as these were not from the group treated with neoadjuvant anti-PD-1. If we remove these two samples, the percentage of large clones in the PBMC and TIL will be similar. Of course, we are aware of the coverage limitation of TRUST4 where we may only observe the largest clones and miss the expansion of the smaller clones both in the TIL and PBMCs. This caveat was stated in the Discussion.

2. I believe Figure 6 and 7 are extraneous. Figure 6 would be appropriate to include, perhaps largely as supplemental, but Figure 7 is more appropriate for a Review and should be excluded here.

- Based off feedback from other reviewers, we believe that the data presented in Figure 6 augments the scRNAseq analysis. Specifically, the data in Figure 6 is an unbiased examination of all potential receptor ligand interactions from the data, which allowed the nomination of additional axes that can be targeted alongside the PD-1:PD-L1 axis; the hypotheses resulting from this interaction analysis may inspire additional clinical testing to further improve the efficacy of immunotherapy in GBM.
- Figure 7 has now been moved to the supplemental. We believe that given the complexity of the paper that having a proposed model or schematic to summarize the findings may be helpful to the readers. However, given that this is only a

proposed model, we agree that it will be more appropriately placed among the other supplementary figures.

Reviewer #2 (Remarks to the Author):

The authors have performed adequate revisions.

Reviewer #3 (Remarks to the Author):

This has been the third time I have reviewed this manuscript and I have seen the progression of this work to a more elegant and nuanced discussion of how neoadjuvant pembro treatment alters the GBM microenvironment.

The authors responded to all of my queries from the previous two submissions, and because of this, I don't have much more to add in terms of reviewer concerns. I like The addition of potential interactions (CellChat) and implications for future therapies (anti-TIGIT). I am also happy with the de-emphasis on the T-cell cytof data as that was my biggest concern.

Below is a minor concern I have regarding the pseudo time trajectory analysis of DCs. The authors suggest that there are changes in DC subsets in Figure 5. however, in the pseudotime analysis, it appears the majority of DC in the newly diagnosed GBM are "low reads". Could this preponderance of low-read cells influence your analysis? Is it possible you see a more inflammatory DC population in neoPD1 because of the differential quality in the DC data?

- We thank the Reviewer for pointing this important observation. We indeed had the same suspicion that the DC data from the newly diagnosed GBM, which showed a significantly higher fraction of the "low read DC", may have been problematic. However, upon closer inspection, we concluded that was not the case. In particular, the DC data were part of the whole tumor, CD45+ single cell data. Importantly, we did not notice any quality and read coverage difference among 1) other myeloid cells and 2) the lymphoid cells when we compared the newly diagnosed GBM against the recurrent GBM (with and without neo-aPD1). As such, we thought a more plausible explanation is that the microenvironment of the newly diagnosed GBM is less friendly (only) to the dendritic cells in such a way that they are less activated/differentiated and somehow ended up in the "low reads" cluster. However, we decided not to speculate about this population because we did not have any evidence that these DCs were indeed defective.